# Repellency and toxicity of long-lasting insecticide-treated bed nets (LLINs) to bed bugs

**Christopher C. Hayes**[ID]*, **Coby Schal**[ID]*

Department of Entomology and Plant Pathology, North Carolina State University, Raleigh, North Carolina, United States of America

* cchayes@ncsu.edu (CCH); coby@ncsu.edu (CS)

## Abstract

Vector control is essential for eliminating malaria, a vector-borne parasitic disease responsible for over half a million deaths annually. Success of vector control programs hinges on community acceptance of products like long-lasting insecticide-treated nets (LLINs). Communities in malaria-endemic regions often link LLIN efficacy to their ability to control indoor pests such as bed bugs (*Cimex lectularius* L. and *Cimex hemipterus* (F.)) (Hemiptera: Cimicidae). Despite this, little is known about the potential repellent effects and toxicity of LLINs to bed bugs. Herein, we demonstrate for the first time that commonly deployed LLINs lack olfactory and contact-based repellency to host-seeking *C. lectularius* from both insecticide-susceptible and insecticide-resistant populations. One LLIN (PermaNet Dual) was significantly attractive to both populations when exposed olfactorily, but not in contact assays, highlighting the complexity of bed bug-LLIN interactions. The insecticide resistant bed bugs experienced low mortality in 4 d of continuous exposure to LLINs. These results suggest that LLINs would likely not repel or eliminate bed bug infestations in malaria-endemic communities, further selecting for insecticide resistance and potentially disrupting vector control programs.

## Introduction

Malaria, a parasitic disease caused by infection with *Plasmodium* (Apicomplexa: Plasmodiidae) and vectored to humans through the bite of infected *Anopheles* mosquitoes (Diptera: Culicidae), is responsible for nearly half a million deaths annually, mainly in sub-Saharan Africa [1]. This immense burden makes malaria control and elimination a global priority [2]. To date, most of the averted disease has been achieved by intervention efforts in two broad areas: 1) treatment of the disease, and 2) disruption of contact between the mosquito vector and its human hosts [3–5]. The use of Long-Lasting Insecticidal Nets (LLINs) has been the single most effective malaria prevention strategy to-date [6]. However, the use of LLINs indoors, particularly on beds, places them in the habitat of bed bugs, *Cimex lectularius* L. and *Cimex hemipterus* (F.) (Hemiptera: Cimicidae) [7]. Both species have resurged globally in the past two

**Data Availability Statement:** All data files are available as a Supplementary Excel file.

**Funding:** This research was funded in part by grants from the Deployed Warfighter Protection Research Program, Department of the Army, U.S.

Army Contracting Command, Aberdeen Proving
Ground, Natick Contracting Division, Ft. Detrick MD
(W911QY1910011), the U.S. Department of
Housing and Urban Development Healthy Homes
program (NCHHU0053-19), a Southern Region
IPM Program grant (416682), and the Blanton J.
Whitmire Endowment. The funders had no role in
study design, data collection and analysis, decision
to publish, or preparation of the manuscript.

**Competing interests:** The authors have declared
that no competing interests exist.

decades in large part due to the widespread emergence of insecticide resistance [8,9] and infestations of these obligate human ectoparasites have become prolific, persistent, and difficult to control.

Malaria-endemic communities that consider the use of LLINs often prioritize the immediate benefits of indoor pest control together with or over the long-term health benefits of mitigating malaria transmission [10–12]. Thus, failure to control indoor pests, such as bed bugs, can contribute to LLIN abandonment, misuse, and ultimately lower efficacy of indoor vector control programs [13,14]. In a recent review of these interactions [7], we highlighted that LLINs can impose strong selection pressure on bed bugs because of the bed bugs' low mobility, tendency to bite their sleeping human host at night, and high affinity for aggregation sites near but not on the host [15–17]. These behaviors place bed bugs in frequent contact with LLINs [18].

Few studies have reported on the behavioral and ecological interactions of bed bugs with insecticide treated bed nets. Recently, we showed that multiple life stages of insecticide-susceptible and insecticide-resistant *C. lectularius* were able to pass through pyrethroid-treated LLINs both in pursuit of a host and when returning to aggregation sites, with minimal mortality only in the insecticide-susceptible bed bugs due to brief interactions with the nets [18]. In these assays we saw no evidence of LLIN repellency of bed bugs. Because these were largely end-point assays that did not track behavior, it remains unknown whether LLINs repel bed bugs. Therefore, we validated the use of a two-choice olfactometer for repellency assays, and then demonstrated its utility for the analysis of repellency in bed bugs using the "gold standard" repellent, N,N-diethyl-meta-toluamide (DEET) [19]. Using this bioassay, herein we evaluated the olfactory and olfactory plus contact repellency of commonly used LLINs, as well as bed bug mortality associated with continuous exposure to LLINs. This investigation was conducted with a highly insecticide-susceptible reference population and a highly insecticide-resistant population to explore the possibility that resistance to insecticides might be accompanied by altered chemosensory sensitivity of bed bugs to LLINs.

## Materials and methods

### Colony maintenance and feeding

Two populations of *C. lectularius* were used in this study. The Harold Harlan population (HH), also known as Ft. Dix, is a commonly used insecticide susceptible reference strain collected at Fort Dix, New Jersey (USA) in 1973 and has not been challenged with insecticides since collection. It was maintained on a human host until December 2008, and then, in our lab, on defibrinated rabbit blood until July 2021 and on human blood thereafter. The Fuller Mill Road population (FM) is resistant to multiple classes of insecticides and was collected from a residence in High Point, North Carolina (USA) in 2017, maintained in our lab on defibrinated rabbit blood until July 2021 and on human blood thereafter [20]. Both populations were maintained in an incubator at 35–45% relative humidity, 25°C on a 12:12 (L:D) h photoperiod and fed weekly on heparinized human blood (supplied by the American Red Cross under American Red Cross IRB #00000288 and protocol #2018–026) using a previously described feeding system [18,21].

Only adult females were used in all repellency assays due to their need to obtain a blood meal between each oviposition cycle and thus high motivation to orient towards human odor. Groups of 20–30 females were separated from colony jars within 48 h post-feeding and starved for 10–14 days at 35–45% relative humidity, 25°C on a 12:12 (L:D) h photoperiod. Since the females were of unknown ages and likely mated within the colony, at several points during this period, but not within 24 h of the assay, groups of 20–30 females were moved onto clean folder

paper in clean 20 ml clear scintillation vials (DWK Life Sciences, Millville, NJ, USA) to remove all eggs. Each bed bug was used for a single bioassay, then discarded. Adult male bed bugs were used in survival assays 4–5 days after they blood-fed.

## Human odor preparation and IRB approval

Human odor samples were collected following a previously validated SOP for human skin swab collection [22,23]. IRB approval for recruitment, with written informed consent, and odor sample collections, was granted from North Carolina State University, Raleigh, NC (IRB Approval #14173). For this study, all samples were obtained from the primary researcher (CCH) between 3 June 2023 and 2 August 2024. Briefly, no alcohol, spicy, or pungent foods were consumed by the subject for at least 24 h prior to collection. Approximately 2–10 h before odor collection, CCH showered using Cetaphil ultra gentle body wash (Galderma, Fort Worth, TX) and no shampoo. No deodorant was applied, and no strenuous activities performed. Hourly, between 2–10 h after showering, CCH cleansed his hands with water (no soap) and once dried, a filter paper (#1, 90 mm diameter, Whatman, Maidstone, United Kingdom) was used to swab a single forearm from wrist to elbow, armpit, and leg from ankle to knee for 30 s per side. A second filter paper was used to swab the same regions on the other side of his body. Each filter paper was cut into 16 equal pie-shaped pieces (4 $cm^2$ each) stored in a glass vial at -20°C and used within one month.

## Bed net sample preparation

We used 5 different bed nets, including 4 different LLINs and an untreated control bed net. These were (1) Siam Dutch (SD), an untreated bed net (Siam Dutch Mosquito Netting Co., Bangkok, Thailand); (2) Olyset Net (OY), containing 800 mg permethrin/$m^2$ (Sumitomo Chemical, Osaka, Japan); (3) PermaNet 2.0 (PN 2.0), containing 56 mg deltamethrin/$m^2$ (Vestergaard, Lausanne, Switzerland); (4) PermaNet 3.0 (PN 3.0), containing two distinct treatments (side panel (S) and roof panel (R)) which were assessed separately and are henceforth referred to as PermaNet 3.0S (PN 3.0S) which contains 84 mg deltamethrin/$m^2$, and PermaNet 3.0R (PN 3.0R) which contains 120 mg deltamethrin/$m^2$ plus 800 mg piperonyl butoxide (PBO)/$m^2$ (Vestergaard); and (5) PermaNet Dual (PND), containing 84 mg deltamethrin/$m^2$ plus 200 mg chlorfenapyr/$m^2$ (Vestergaard). The SD and OY nets were provided as new nets by the U.S. Centers for Disease Control (CDC, Atlanta, GA). Cuttings of newly manufactured PermaNets from two distinct production runs were provided by Vestergaard. Individual squares (15 cm x 15 cm; 0.0225 $m^2$) of each bed net were prepared for olfaction-only repellency assays, and rectangles (1.5 cm x 3.0 cm; 0.00045 $m^2$) for olfaction plus contact assays. Squares used in olfaction assays were replaced weekly, and rectangles used in olfaction plus contact assays were replaced between each assay. Individual squares (8 cm x 8 cm; 0.0064 $m^2$) of each bed net were prepared for survival assays, for each replicate, and then discarded. Nitrile gloves (Layer4, USA Scientific, Ocala, FL) were worn in all manipulations of bed nets, chemicals, olfactometer components, and bed bugs, and they were replaced regularly to prevent cross-contamination and transfer of human skin compounds.

## Chlorfenapyr dispenser preparation

Individual squares (1.5 cm x 1.5 cm; 0.000225 $m^2$) of #1 Whatman filter papers were placed on aluminum foil, treated with either 10 μl of 99.9% acetone (Sigma-Aldrich, Burlington, MA) (control) or 10 μl of 98.0% chlorfenapyr (PESTANAL analytical standard, Fisher Scientific, Waltham, MA) in acetone. Papers were then allowed to air dry for at least 20 min and stored

in glass vials (20 ml) at -20˚C. Concentrations of chlorfenapyr tested were 0.1, 1.0, and 10.0 µg/µl, representing total doses of 1, 10, and 100 µg, respectively, applied to each filter paper.

## Olfactometer assays

All assays were performed using a glass Y-tube olfactometer [22,23] and methodology as previously described [19], with minor modifications. Briefly, a vertically oriented Y-tube was connected to a forced air system (Fig 1). A plankton mesh (Wildco, Yulee, FL) walkway was used to facilitate bed bugs crawling on a vertical surface, and it was replaced after no more than five replicate assays or for each assayed bed net, whichever came first.

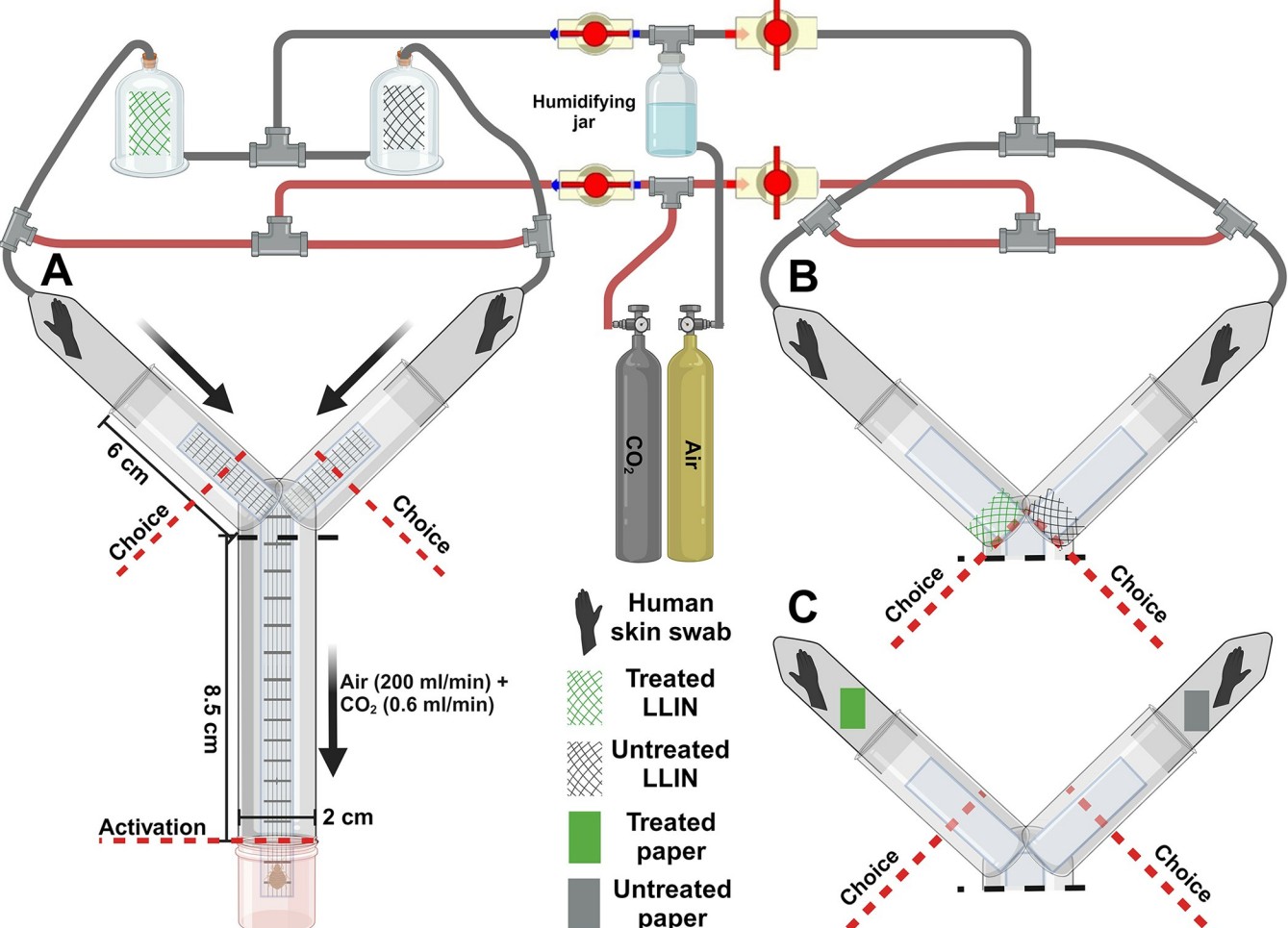

**Fig 1. Schematic of the binary choice olfactometer used to quantify the responses of *C. lectularius* to four different LLINs and to technical chlorfenapyr.** Assays tested olfactory repellency without contact exposure (**A**, **C**) and the combination of olfactory and contact repellency (**B**). In all assays, host-seeking adult female bed bugs were attracted toward human-associated olfactory cues (skin volatiles and $CO_2$) delivered in humidified air. Glass jars containing cuttings of LLINs served to deliver LLIN odors in olfaction-only assays (**A**), but they were not used in olfaction plus contact assays (**B**) and chlorfenapyr olfaction assays (**C**). In this schematic, the olfaction-only assay (**A**) is depicted with the air and $CO_2$ lines leading to **B** closed. When assays **B** or **C** were used, the air and $CO_2$ lines leading to **B** or **C** were open and the lines leading to **A** were closed. Behavioral responses to various LLINs and chlorfenapyr were quantified by measuring percentage Activation (bed bugs that entered the olfactometer / total bed bugs assayed); percentage Choice (bed bugs that made a choice of either arm of the olfactometer / total bed bugs that activated); and percentage Preference (percentage that chose the treatment vs. control arms of the olfactometer). LLINs, chlorfenapyr-treated filter papers, and human skin swab stimuli were prepared independently and introduced separately. Human skin swabs were replaced after each replicate, and the walkway was changed after either 5 replicates or between treatment groups, whichever came first. Created with BioRender.com.

In olfaction-only LLIN assays (Fig 1A), medical quality humidified air (200 ml/min) (Airgas Healthcare, Radnor, PA) was split so 100 ml/min passed through each glass jar (8 cm x 12.5 cm; ~628.3 $cm^3$) (Prism Glass, Raleigh, NC) that contained a square cutting of a bed net, rotating between samples from different production runs when possible (PermaNet samples), each day of data collection. A square cutting of an untreated bed net (SD) was placed in one of the bed net jars and one of the four LLINs was placed in the other. Then, $CO_2$ (0.3 ml/min, ~3,000 ppm, Airgas Healthcare, Radnor, PA) was added to each air stream before it entered the olfactometer. At the distal end of each arm of the olfactometer we placed a human skin swab. In olfaction plus contact LLIN assays (Fig 1B), all parameters were identical to the olfaction-only assay, except that odor jars containing LLIN cuttings were removed, and a rectangular cutting of bed net was affixed to the walkway at the proximal end of one arm of the olfactometer, rotating between samples from different production runs for all PermaNet samples each day of data collection. A cutting of the untreated SD bed net was affixed to the walkway of the other arm. In the olfaction-only assays with unformulated technical grade chlorfenapyr, all parameters were identical to the LLIN olfaction assays, except that the LLIN jars shown in Fig 1A were removed. A chlorfenapyr-treated filter paper, rather than LLIN cutting, was placed at the distal end of one arm of the olfactometer and an acetone-treated filter paper was placed in the other arm, each adjacent to a skin swab filter paper, but preventing contact of the two treated papers (Fig 1C).

In all assays, individual bed bugs were acclimatized to 200 ml/min of air for at least 30 min in separate releasing tubes, and then introduced to the assay via the uncapped releasing tube. Activation (moving from the releasing tube into the common arm of the olfactometer), Choice (moving more than halfway up either of the assay arms (Fig 1A and 1C) or crossing onto either the untreated or treated bed net affixed to the walkway (Fig 1B)), and Preference (selected assay arm) were recorded up to 5 min or to when a choice was made, whichever came first.

## Bed bug mortality assays

All mortality assays were performed in inverted plastic jars (5.5 cm × 4.8 cm each; Olcott Plastics, Saint Charles, IL) with the bottom of each jar and the center of each lid removed. Square cuttings of each bed net were affixed to the top of each jar and secured using the ring-shaped lid. Jars were then inverted, and groups of 10 adult male bed bugs were placed on each bed net, for both HH and FM populations, with all LLINs run in triplicate ($n$ = 30 bed bugs per LLIN per population). Bed bugs remained in continuous contact with the bed net surface throughout the assay, and mortality was scored at 0.25, 0.5, 1, 2, 3, 6, 8, 12 h and then every 12 h until 96 h. Bed bugs were considered dead if they failed to move when gently touched using feather light forceps and they were subsequently unable to right themselves when flipped onto their dorsal side.

## Statistical analysis

All statistical analyses were conducted using SAS Enterprise Guide (v. 8.3, SAS Institute, Cary, NC), with α = 0.05. Percentage activation and choice data were first arcsine square-root-transformed, and then compared via one-way ANOVA within each population and generalized linear model (GLM) followed by Tukey's HSD between the two populations. Within populations, LLIN-specific percentage preference was compared using individual Chi-square tests. Comparison of proportion surviving on each LLIN over time was done using Kaplan-Meier survival analysis and log-rank tests with Sidak correction for multiple comparisons.

## Results

### Lack of olfactory repellency of LLINs

Using the olfaction-only bed net assay (Fig 1A) we evaluated the repellency of five bed net treatments representing four commonly distributed LLINs, as well as an untreated control bed net. High percentage activation (91.7%) and choice (86.4%) were seen with the HH bed bugs in the positive controls (human odor and $CO_2$ only in one arm of the olfactometer) (Fig 2A). Similarly, in assays run with the FM bed bugs, both positive controls and concurrently run HH bed bugs as controls (HC: run daily alongside FM controls to allow for cross-population comparisons) showed high activation (95.2%, 91.3%) and choice (90.0%, 85.0%), respectively (Fig 2B). In the HH, FM, and HC controls, 100% of the bugs that made a choice preferred the human odor-containing arm of the Y-tube (Chi-square test, HH: $\chi^2 = 19$, $df = 1$, $P < 0.01$; FM: $\chi^2 = 18$, $df = 1$, $P < 0.01$; HC: $\chi^2 = 18$, $df = 1$, $P < 0.01$). We validated the untreated SD net as a negative control (NC) by placing a square cutting on one side of the olfactometer with human

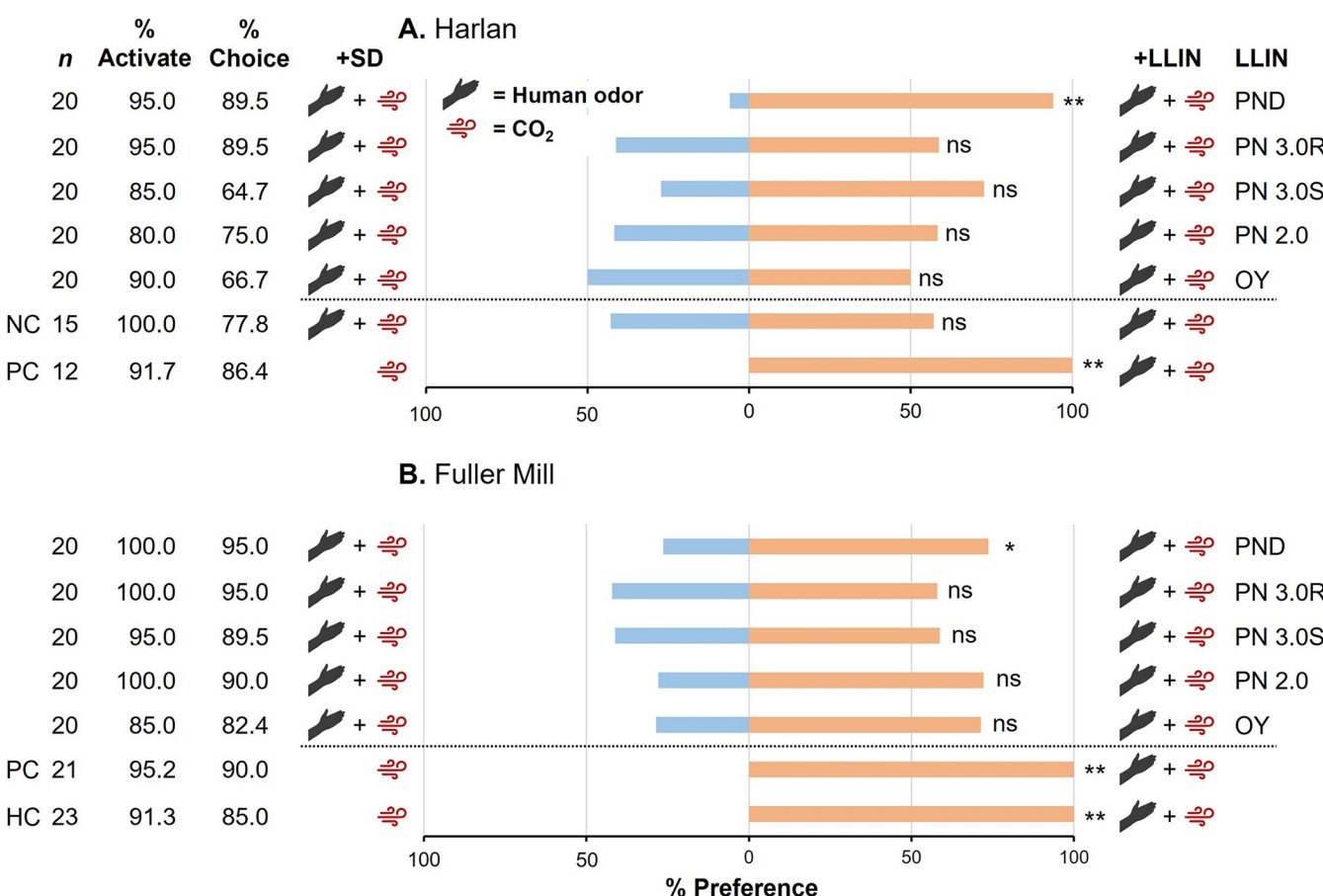

**Fig 2.** Comparisons of olfaction-mediated behavioral responses of insecticide-susceptible (Harold Harlan) (A) and insecticide-resistant (Fuller Mill) (B) *C. lectularius* to various LLINs. Individual adult females were assayed 10–14 d post blood meal. The positive control (PC) and the Harold Harlan control (HC in **B**; run daily alongside Fuller Mill PC) consisted of host cues (human odor and/or $CO_2$) at only one arm of the olfactometer. The net negative control (NC in **A**) had identical host cues at both arms of the olfactometer. The percentage choosing the subject stimuli, including LLIN (right side, orange), and towards the control stimuli, including the untreated Siam Dutch (SD) net (left side, blue) are shown. Percentage Activate was compared within each graph (one-way ANOVA) and the overall model for each was not significant ($P > 0.05$). Likewise, % Choice was compared within each graph (one-way ANOVA) and between bed bug populations (GLM), and both sets of comparisons did not reveal any significant differences ($P > 0.05$). Preference was compared independently for each LLIN by Chi-square test, with asterisks representing significant differences in preference denoted by *, $P < 0.05$; and **, $P < 0.01$.

odor and $CO_2$ supplied to both sides, revealing similarly high activation (100%) and choice (77.8%), and no significant preference for the arm containing the control SD net ($\chi^2 = 0.3$, $df = 1$, $P > 0.05$).

The HH bed bugs showed 80.0–95.0% activation across all LLIN assays (Fig 2A), with no significant change in activation across tested LLINs (one-way ANOVA, $F = 1.12$, $df = 6,27$, $P = 0.3758$). Similarly, 85.0–100% activation was seen across all LLIN assays with the FM bed bugs (Fig 2B), and no significant change in activation across LLINs (one-way ANOVA, $F = 1.17$, $df = 5,23$, $P = 0.3692$). Comparison of activation between HH and FM bed bugs revealed an overall nonsignificant model (GLM, $F = 1.14$, $df = 12,59$, $P = 0.3505$) with neither population ($F = 3.61$, $df = 1$, $P = 0.0622$), LLIN ($F = 1.38$, $df = 6$, $P = 0.2388$), or the interaction of population and LLIN ($F = 0.89$, $df = 5$, $P = 0.4924$) significantly affecting percentage activation. Percentage choice was also high in both HH (64.7–89.5%) (Fig 2A) and FM (82.4–95.0%) bed bugs (Fig 2B), with no significant decrease in choice associated with LLIN in either population (one-way ANOVA, HH: $F = 0.87$, $df = 6,27$, $P = 0.5279$; FM: $F = 0.69$, $df = 5,23$, $P = 0.6355$). Comparison of percentage choice between HH and FM bed bugs revealed an overall nonsignificant model (GLM, $F = 1.17$, $df = 12,59$, $P = 0.3287$) with neither population ($F = 3.87$, $df = 1$, $P = 0.0538$), treatment ($F = 1.51$, $df = 6$, $P = 0.1898$), or the interaction of population and treatment ($F = 0.17$, $df = 5$, $P = 0.9719$) significantly affecting percentage choice.

Comparison of percent preference across treatments revealed unexpected results in both bed bug populations. In both HH and FM bed bugs we saw no significant attraction or repellency when exposed to any of the single-ingredient or synergist-containing LLINs (Chi-square tests, OY: HH–$\chi^2 = 0.0$, $df = 1$, $P > 0.05$, FM–$\chi^2 = 2.57$, $df = 1$, $P > 0.05$; PN 2.0: HH–$\chi^2 = 0.33$, $df = 1$, $P > 0.05$, FM–$\chi^2 = 3.6$, $df = 1$, $P > 0.05$; PN 3.0S: HH–$\chi^2 = 2.2$, $df = 1$, $P > 0.05$, FM–$\chi^2 = 0.53$, $df = 1$, $P > 0.05$; PN 3.0R: HH–$\chi^2 = 0.53$, $df = 1$, $P > 0.05$, FM–$\chi^2 = 0.47$, $df = 1$, $P > 0.05$) (Fig 2). However, there was surprisingly significant attraction to the chlorfenapyr-containing PND net in both the HH bed bugs ($\chi^2 = 13.24$, $df = 1$, $P < 0.01$), and to a lesser degree in FM bed bugs ($\chi^2 = 4.26$, $df = 1$, $P < 0.05$). The results with PND focused our attention on the statistically nonsignificant trend of bed bugs from both populations toward attraction to the other three LLINs (OY, PN 2.0, and PN 3.0 (both side and top panels)). To increase the power of the analysis, we combined all these assays in a single Chi-square analysis ($n = 80$ assays per population). In the case of HH, we still did not detect any significant preference ($\chi^2 = 1.923$, $df = 1$, $P > 0.05$). However, in the case of FM bed bugs, we saw overall significant attraction to LLINs when treatments were combined ($\chi^2 = 5.882$, $df = 1$, $P < 0.05$). These results suggest that while each LLIN did not repel insecticide-susceptible and resistant bed bugs, there was a tendency to orient toward the bed nets, particularly for the FM bed bugs. However, the PND LLIN was the only individual treatment that significantly attracted bed bugs of both populations.

## Marginal repellency of LLINs in contact assays

To further investigate the potential attractiveness of the PND LLIN, and to introduce contact with LLINs as an additional sensory modality, we assessed repellency in response to both olfactory and contact stimuli in the same representative bed nets (Fig 1B). Once again, we saw 93.3% activation and 85.7% choice in the positive controls with HH bed bugs (Fig 3A), as well as in HH bugs concurrently run with FM bed bugs (HC controls, 100% and 90.9%, respectively) (Fig 3B). All FM positive control bed bugs activated and made a choice (Fig 3B). In all HH, FM, and HC controls, 100% of the bed bugs that made a choice preferred the human odor-containing arm of the Y-tube (Chi-square test, HH: $\chi^2 = 12$, $df = 1$, $P < 0.01$; FM: $\chi^2 = 10$, $df = 1$, $P < 0.01$; HC: $\chi^2 = 10$, $df = 1$, $P < 0.01$).

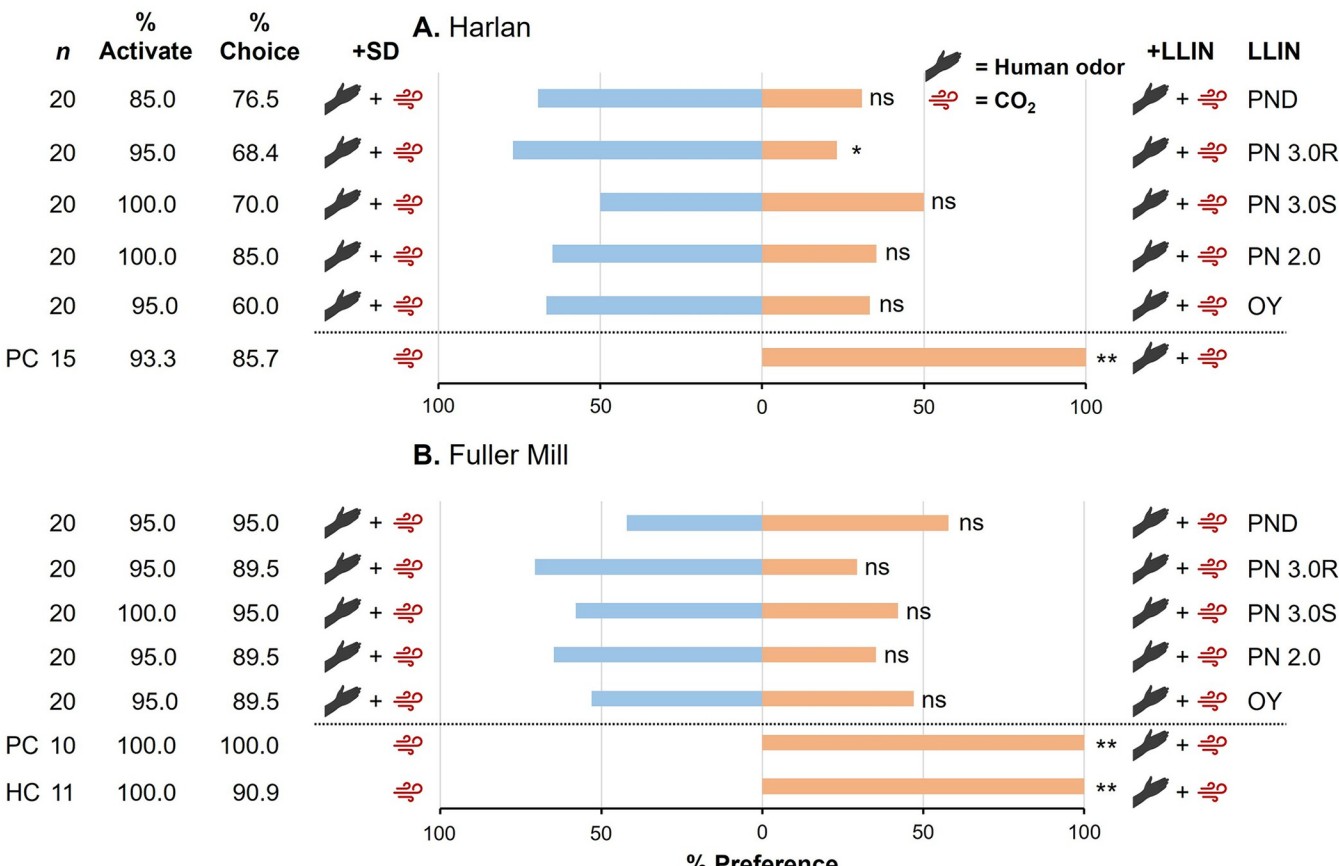

**Fig 3.** Comparisons of olfaction plus contact-mediated behavioral responses of insecticide-susceptible (Harold Harlan) (A) and insecticide-resistant (Fuller Mill) (B) *C. lectularius* to various LLINs. Individual adult females were assayed 10–14 d after ingesting a blood meal. The positive control (PC) and the Harold Harlan control (HC in **B;** run daily alongside Fuller Mill PC) consisted of host cues (human skin odor and/or $CO_2$) at only one arm of the olfactometer. The percentage preference toward subject stimuli, including LLIN (right side, orange), and towards the control stimuli, including the untreated Siam Dutch (SD) net (left side, blue) are shown. Percentage Activate was compared within each graph (one-way ANOVA) and the overall model for each was not significant ($P > 0.05$). Likewise, % Choice was compared within each graph (one-way ANOVA) and between the two bed bug populations (GLM), and both sets of comparisons did not reveal any significant differences ($P > 0.05$). Preference was compared independently for each LLIN by Chi-square test, with asterisks representing significant differences in preference denoted by *, $P < 0.05$; and **, $P < 0.01$.

The olfaction plus contact assays also resulted in high overall activation in both the HH (85.0–100%) and the FM (95.0–100%) bed bugs, with no significant decrease in activation based on LLIN for either population (one-way ANOVA, HH: $F = 0.89$, $df = 5,20$, $P = 0.5039$; FM: $F = 0.46$, $df = 5,19$, $P = 0.8038$). Comparison of percentage activation across both populations revealed an overall nonsignificant model (GLM, $F = 0.76$, $df = 11,44$, $P = 0.6740$), with neither population ($F = 0.12$, $df = 1$, $P = 0.7339$), LLIN ($F = 1.22$, $df = 5$, $P = 0.3156$), or the interaction of population and LLIN ($F = 0.47$, $df = 5$, $P = 0.7948$) significantly affecting activation. Further, percentage choice remained high across both populations (HH: 60.0–85.0%; FM: 89.5–95.0%) regardless of LLIN, with no significant decrease in percentage choice associated with LLIN in either HH or FM bed bugs (one-way ANOVA, HH: $F = 2.11$, $df = 5,25$, $P = 0.0979$; FM: $F = 1.02$, $df = 5,19$, $P = 0.4361$) (Fig 3). Comparison between the two populations of the percentage bed bugs that made a choice revealed an overall significant model (GLM, $F = 2.46$, $df = 11,44$, $P = 0.0170$) with population significantly affecting choice ($F = 14.56$, $df = 1$, $P = 0.0004$), but neither LLIN ($F = 2.36$, $df = 5$, $P = 0.0557$) nor the

interaction of population and LLIN ($F = 0.53$, $df = 5$, $P = 0.7494$) significantly affecting percentage choice.

Comparison of percentage preference across treatments following the addition of contact exposure in the HH population revealed no significant attraction or repellency when exposed to three of the four LLINs (OY: $\chi^2 = 1.3$, $df = 1$, $P > 0.05$; PN 2.0: $\chi^2 = 1.5$, $df = 1$, $P > 0.05$; PN 3.0S: $\chi^2 = 0.0$, $df = 1$, $P > 0.05$). However, PN 3.0R was significantly repellent to the HH bed bugs ($\chi^2 = 5.4$, $df = 1$, $P < 0.05$) (Fig 3A). In the FM bed bugs we saw no significant attraction or repellency when bed bugs were exposed to any of the single-ingredient or synergist-containing LLINs (OY: $\chi^2 = 0.06$, $df = 1$, $P > 0.05$; PN 2.0: $\chi^2 = 1.5$, $df = 1$, $P > 0.05$; PN 3.0S: $\chi^2 = 0.47$, $df = 1$, $P > 0.05$; PN 3.0R: $\chi^2 = 2.9$, $df = 1$, $P > 0.05$) (Fig 3B). The previously observed attraction to the PND LLIN was eliminated in both populations upon the addition of contact exposure (HH: $\chi^2 = 1.9$, $df = 1$, $P > 0.05$; FM: $\chi^2 = 0.47$, $df = 1$, $P > 0.05$). Thus, both single- and multi-AI-impregnated LLINs (except for PN 3.0R and HH strain) did not significantly repel host-seeking bed bugs.

## No olfactory attraction or repellency to chlorfenapyr

We sought to understand why the PND bed net was attractive to bed bugs. Because the other deltamethrin-containing LLINs (PN 2.0, 3.0S, 3.0R) were not individually attractive to bed bugs, we tested whether chlorfenapyr might attract bed bugs (Fig 1C). In the presence of human odor and $CO_2$ 100% of the bed bugs were activated and 90.0% made a choice ($\chi^2 = 11$, $df = 1$, $P < 0.01$) (Fig 4). Also, in the presence of technical chlorfenapyr, 85.0–95.0% of the bed bugs activated with no significant differences across the tested doses of 1, 10 and 100 µg (one-way ANOVA, $F = 2.11$, $df = 3,12$, $P = 0.1522$). Similarly, high percentages of the bed bugs made a choice across all tested doses (76.5–84.2%) with no significant effect of chlorfenapyr dose (one-way ANOVA, $F = 0.62$, $df = 3,12$, $P = 0.6176$). Finally, there was no significant preference across doses of chlorfenapyr (1 µg: $\chi^2 = 0.5$, $df = 1$, $P > 0.05$; 10 µg: $\chi^2 = 1.8$, $df = 1$, $P > 0.05$; 100 µg: $\chi^2 = 3.77$, $df = 1$, $P > 0.05$), suggesting no significant repellency of chlorfenapyr. However, a trend across the three doses suggests that higher doses of chlorfenapyr might repel HH bed bugs (Fig 4).

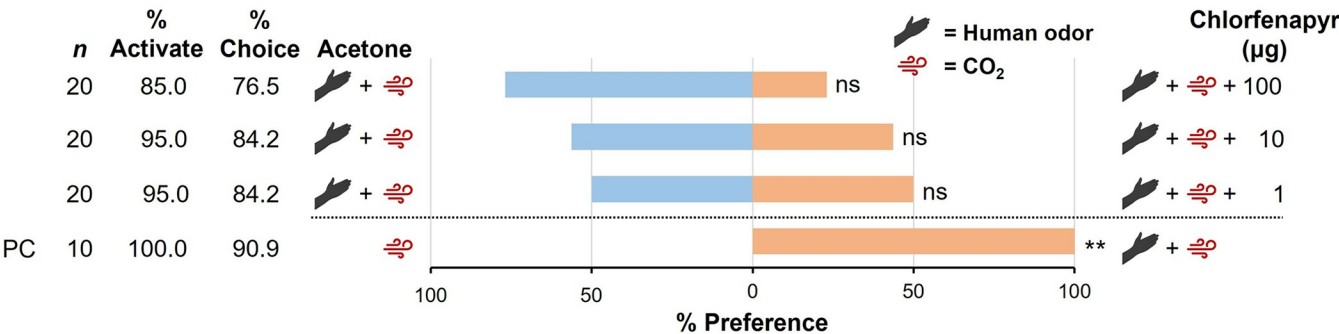

**Fig 4. Comparison of olfaction-mediated behavioral responses of insecticide-susceptible (Harold Harlan) *C. lectularius* to various doses of chlorfenapyr.** Individual adult females were assayed 10–14 d after a blood meal. The positive control (PC) consisted of host cues (human skin odor and/or $CO_2$) at only one arm of the olfactometer. The percentage Preference toward subject stimuli, including chlorfenapyr dose (right side, orange), and towards the control stimuli, including acetone-treated paper (left side, blue) are shown. Percentage Activate and % Choice were compared separately (one-way ANOVA) and the overall model for each was not significant ($P > 0.05$). Preference was compared independently at each dose by Chi-square test, with asterisks representing significant differences in preference denoted by *, $P < 0.05$; and **, $P < 0.01$.

## Bed bug mortality on LLINs

Because the LLINs were not repellent to either of the two tested bed bug populations in olfactory (Fig 2) or olfactory plus contact assays (Fig 3), we surmised that bed bugs might frequently contact the LLINs and possibly harbor in the creases of the LLINs under field conditions. Therefore, we exposed 180 bed bugs of each strain to five LLINs treatments in 4-day long continuous contact assays. The HH bed bugs reached 100% mortality across all LLINs within 12 h, whereas 100% of the bed bugs survived on the SD untreated bed nets for 96 h (shown to 16 h in Fig 5A). The resistant FM bed bugs reached only 80% mortality in a single LLIN treatment (PN 3.0R: deltamethrin + PBO) by 96 h, with 100% survival on the SD controls (Fig 5B). Comparison of population-specific survival revealed significantly lower proportion surviving on LLINs in both the HH (Kaplan-Meier, log-rank, $\chi^2$ = 261.9, $df$ = 1, $P$ < 0.0001) and FM ($\chi^2$ = 46.0, $df$ = 1, $P$ < 0.0001) populations (Fig 5) compared to the untreated SD control. Comparison of the two bed bug populations by LLIN revealed that significantly higher proportions of FM bed bugs than HH bed bugs survived on all the LLINs. Analysis of survival over-time revealed overall similar survival time in both HH and FM bed bugs regardless of LLIN, with PN 3.0R causing the most rapid mortality (log-rank, HH: $\chi^2$ = 175.8, $df$ = 1, $P$ < 0.0001; FM: $\chi^2$ = 41.9, $df$ = 1, $P$ < 0.0001). These results demonstrate that, with long uninterrupted LLIN exposure representing sheltering by blood-fed bed bugs, only a fraction of the insecticide-resistant bed bugs (FM strain) died (20–67% survival), suggesting that LLINs can impose strong selection pressure for the emergence and maintenance of insecticide resistance in bed bug populations.

## Discussion

To our knowledge, this is the first study to 1) compare the olfactory and contact repellency of commonly used LLINs to both insecticide-susceptible and insecticide-resistant bed bugs, 2) assess the survival of bed bugs over time with continuous exposure to multiple LLINs, and unexpectedly 3) present empirical data showing behavioral olfactory attraction of bed bugs to an insecticide-impregnated fabric, and loss of attraction following the addition of contact exposure to the LLIN. We were able to show this through the use of a robust binary choice olfactometer system that was previously validated to assess repellency in bed bugs [19]. Briefly, we have shown that olfactory exposure to commonly distributed single- and multi-ingredient LLINs did not repel either insecticide-susceptible or insecticide-resistant *C. lectularius* (Fig 2). Further, the addition of contact exposure to the same LLINs also did not repel bed bugs of both populations, except for the HH bed bugs exposed to PN 3.0R (deltamethrin + PBO) (Fig 3A).

### Repellency of insecticides to bed bugs

Our understanding of insecticide repellency to bed bugs is lacking, and even more so as it relates to the association between insecticide resistance and changes in repellency. Different insecticide resistant bed bug populations appear to exhibit differing responses to DEET, some showing lower and others expressing higher sensitivity to DEET [19,24], consistent with findings in other insect species [25–28]. Of particular interest is repellency of bed bugs to pyrethroids, the most widely used class of insecticides. Here too, both repellency and lack of repellency have been reported for the same pyrethroid insecticides [29–32]. Notably, different repellency assays might account for disparate results, including the use of olfactory- or contact-based assays and observational vs. endpoint assays.

Herein, we have provided clear evidence that LLINs containing pyrethroids, irrespective of exposure modality, generally do not repel host-seeking, insecticide-susceptible and

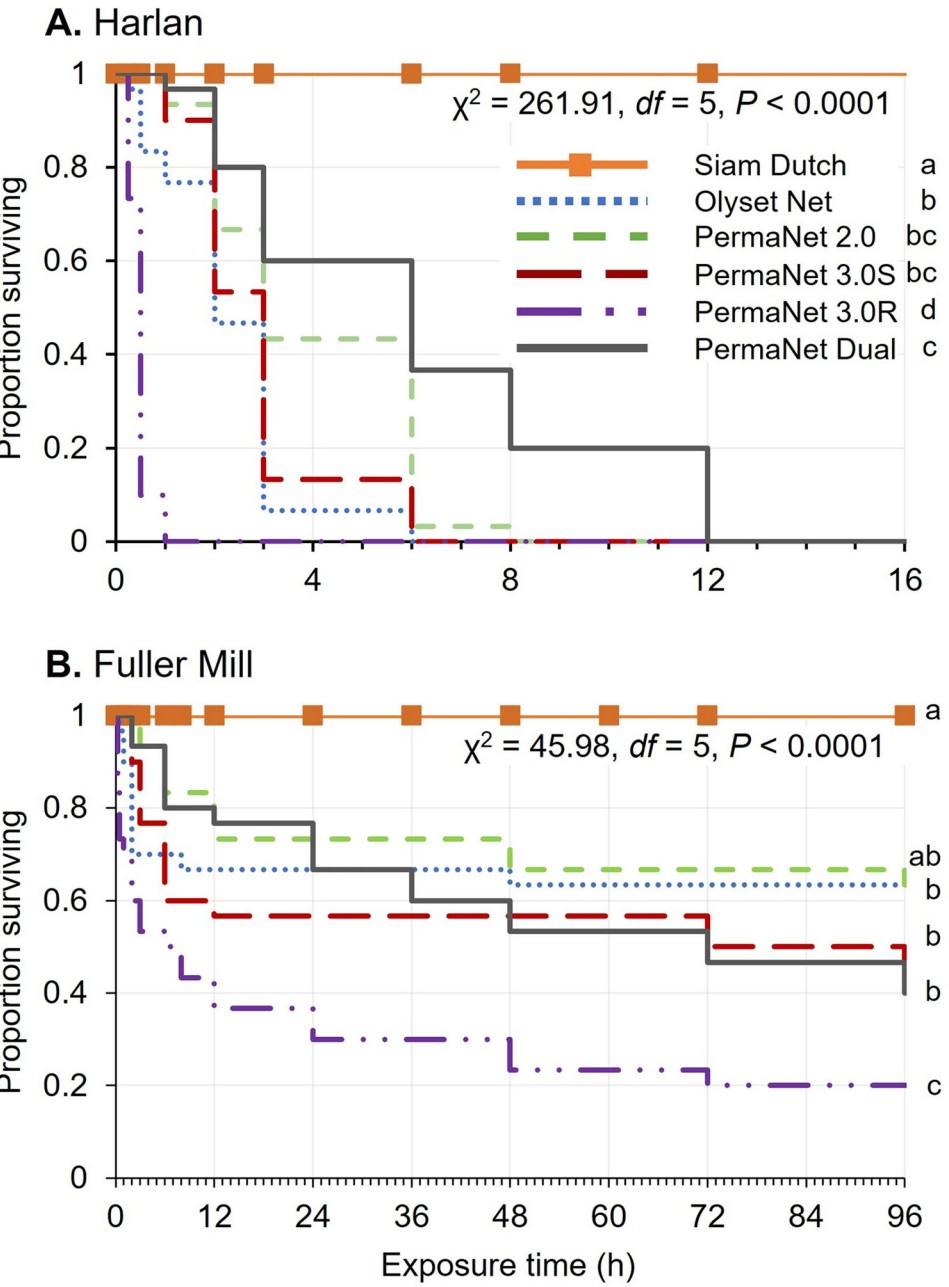

**Fig 5. Comparison of insecticide-susceptible (Harold Harlan) (A) and insecticide-resistant (Fuller Mill) (B) *C. lectularius* surviving up to 96 h continuous contact with LLINs.** For each of three replicates with each LLIN (*n* = 30 bed bugs per LLIN per strain), a group of 10 males, 4 days post-feeding, was placed on a 15.9 cm² circle of LLIN. Mortality was scored at 0.25, 0.5, 1, 2, 3, 6, 8, 12 h and then every 12 h until 96 h. Comparison of proportion surviving across all LLINs was done using Kaplan-Meier tests, and log-rank tests with Sidak correction for multiple comparisons. *P*-values are reported, and strain-specific LLIN comparisons sharing lower-case letters are not significantly different.

insecticide-resistant bed bugs (Figs 2 and 3). Nonetheless, we observed significant repellency of HH bed bugs in olfaction plus contact assays with PN 3.0R, which contains 120 mg/m² deltamethrin plus 800 mg/m² PBO. It is unclear however what feature of this LLIN is repellent to bed bugs. Also, PN 3.0R is the roof panel of the PermaNet 3.0 LLIN, and the likelihood of frequent bed bug contact with the LLIN roof during host-seeking in the field is not known.

Previous research has demonstrated the clear synergistic effects of PBO when used in concert with pyrethroids, both in bed bugs and other insect species [33–36]. Despite its utility as a synergist to overcome insecticide resistance, it has been shown that the application of PBO within vector control settings can disrupt the efficacy of residual insecticides, highlighting its complex nature [37,38]. Importantly, PBO elicits irritancy behaviors in field populations of the mosquito *Aedes aegypti* (L.) (Diptera: Culicidae) [39], suggesting that PBO in PermaNet 3.0 LLIN might be responsible for contact repellency in bed bugs. However, this mechanism might be limited to pyrethroid-susceptible bed bugs, which are rarely found in recent global collections representing hundreds of populations [8]. The increasing reliance on PBO and other synergists to overcome pyrethroid resistance in bed bugs, and its use in LLINs and indoor residual sprays compel more research on its contact repellency to bed bugs and disease vectors as it might lessen their intended contact with residual insecticides.

## Attraction of bed bugs to bed nets

We demonstrated significant olfactory attraction of *C. lectularius* to PermaNet Dual, a dual-AI LLIN with 84 mg/m$^2$ deltamethrin and 200 mg/m$^2$ chlorfenapyr (approximately 4.5 mg of chlorfenapyr in the 0.0225 m$^2$ LLIN cutting used in olfaction assays) (Fig 2). This finding was consistent across two production lots of this bed net. To our knowledge, this is the first documented case of olfactory attraction of an insect to an insecticide-impregnated fabric. It is important to note that while significant olfactory attraction was not seen with any of the other nets, there was an overall trend in preference towards all LLINs in olfactory assays (Fig 2), and an overall trend in preference away from LLINs in contact assays (Fig 3).

Of particular interest in this case is that bed bugs were not attracted to three other tested LLINs that contained deltamethrin (PermaNet 2.0, 56 mg/m$^2$ deltamethrin; the side panel of PermaNet 3.0 (PN 3.0S), 84 mg/m$^2$ deltamethrin; and the roof of PermaNet 3.0 (PN 3.0R), 120 mg/m$^2$ deltamethrin and 800 mg/m$^2$ PBO) (Fig 3), nor to various concentrations of technical grade chlorfenapyr (Fig 4). Therefore, we suspect that neither deltamethrin nor chlorfenapyr contributed to bed bug attraction. However, little is known about the interactions of bed bugs with technical chlorfenapyr, as most of the research with this AI has focused on the efficacy of formulated products [40–43]. Therefore, it would be instructive to conduct more extensive behavioral assays with a wider range of chlorfenapyr concentrations.

Two other possibilities might account for the observed attraction to PermaNet Dual. First, this LLIN might have been contaminated with attractive odors. This is possible in two ways: 1) through handling when packaged immediately following production, or 2) odorant(s) produced as a result of the manufacturing process that result in an attractive response. We suspect that contamination through handling is unlikely because all other PermaNet LLINs were also packaged by the same individual at Vestergaard, and no attraction was observed with those nets (Fig 2). The second possibility is that constituents on this LLIN (possibly chemicals used in production) contributed to bed bug attraction. This too is unlikely because both PermaNet Dual and PermaNet 3.0 are knitted using the same base fabric and impregnated via the same process which includes a water-only rinse after knitting, but no additional washing after the impregnation process (Vestergaard, personal communication), yet we did not see significant attraction to PermaNet 3.0. However, to our knowledge no research exists with any arthropod, comparing the attractiveness of unwashed new LLINs to washed LLINs; therefore, it is unknown if bed bugs and other insects are attracted only to unwashed new LLINs.

Regardless, the WHO has no formal requirement for LLIN washing frequency, and while it has been shown that communities typically wash LLINs monthly, it is likely that new LLINs are used directly out of the packaging without washing and therefore may attract bed

bugs and other arthropods until first washed [44]. Further research in this area is critical, first to independently confirm our finding of attraction of bed bugs to PermaNet Dual, and then experiments to assess the potential attractiveness of LLINs and the effect of washing on potential attractiveness to better understand their impact on insect behavior. The phenomenon of LLINs attracting pest insects presents potential challenges and opportunities in pest management. Namely, when disease vectors, such as *Anopheles* mosquitoes, are found indoors, it might be advantageous to attract them to LLINs to facilitate contact with the insecticide and potentially kill the mosquitoes. However, in the case of flightless and obligatory household pests like bed bugs, attractive LLINs could move them closer to human hosts and more persistent contact with the LLIN might select more rapidly for insecticide resistance.

## Bed bug survivorship on LLINs and vector control

We have performed the first comprehensive time-course of mortality of insecticide-susceptible and insecticide-resistant bed bugs exposed to a variety of LLINs representing field-deployed historic and modern products (Fig 5). Our results demonstrate that while 100% of the insecticide-susceptible HH bed bugs died within 12 h, at most 80% of the insecticide-resistant FM bed bugs died after 4 days of continuous exposure. These findings suggest that while these LLINs are highly effective on mosquitoes, their field efficacy on bed bugs is likely mediocre, highlighting the long-cited concerns of communities in malaria-endemic regions with bed bug infestations [45–49]. The declining effectiveness of LLINs on bed bugs and other household pests, due in large part to the emergence of insecticide resistance, has been shown to decrease community trust and acceptance of LLINs, challenging the efficacy of indoor vector control programs and even leading to program failure [11,50–52]. In a recent review, we summarize these interactions and consider future directions [7].

## Field relevance and limitations

It is important to note the limitations of this study in the contexts of both bed bug behavior and vector control. The Y-tube olfactometer is a robust assay that has been effective in assessing bed bug repellent behaviors. However, due to time constraints associated with assaying individual bed bugs, a final sample size of 20 insects per treatment group was used for this study. This sample size had the power to assess significant changes in bed bug preference to provided stimuli, but with an increased sample size certain results may shift from non-attractive or non-repellent to attractive and repellent, respectively (Figs 2 and 3). Additionally, all assays herein were run using *C. lectularius* bed bugs, which may exhibit different behaviors than those observed in *C. hemipterus* bed bugs when exposed to LLINs. Therefore, similar assays should be run using *C. hemipterus* bed bugs. However, in the context of vector control programs, *C. lectularius* and *C. hemipterus* overlap throughout Africa, with both species facing selection pressures associated with the widespread use of LLINs [53].

Failure to eliminate bed bug populations, together with the impressive ability of small bed bug propagules to withstand the adverse effects of inbreeding and establish large indoor populations, highlight the immense selection pressure imposed on bed bug populations by the widespread use of LLINs. These findings also underscore the need for WHO, NGOs and LLIN manufacturers to consider the adverse effects of bed bug infestations on campaigns to reduce malaria, and conversely, the ancillary benefits of bed bug elimination on LLINs adoption, use, and retention.

## Supporting information

**S1 Data.**
(XLSX)

## Acknowledgments

We thank Drs. Ke Dong, Fred Gould, and Michael Roe for valuable suggestions on an earlier draft of this manuscript. We are grateful to Drs. Tie Lan and Sha Fu for providing all Perma-Net LLINs used in this study, and Dr. Steve Smith (CDC) for providing the Olyset Net and Siam Dutch samples. We appreciate Rick Santangelo's help with colony maintenance and sorting bed bug females and males for this and other projects. CCH also thanks the National Pest Management Association and the NC Pest Management Association for scholarships in support of this research.

## Disclaimer

The content is solely the responsibility of the authors and does not necessarily represent the official views of the U.S. Government and no official endorsement should be inferred.

## Author Contributions

**Conceptualization:** Christopher C. Hayes, Coby Schal.

**Data curation:** Christopher C. Hayes.

**Formal analysis:** Christopher C. Hayes.

**Funding acquisition:** Coby Schal.

**Investigation:** Christopher C. Hayes.

**Methodology:** Christopher C. Hayes.

**Project administration:** Coby Schal.

**Resources:** Coby Schal.

**Supervision:** Coby Schal.

**Validation:** Christopher C. Hayes, Coby Schal.

**Visualization:** Christopher C. Hayes, Coby Schal.

**Writing – original draft:** Christopher C. Hayes.

**Writing – review & editing:** Christopher C. Hayes, Coby Schal.

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
