## [Decision Letter · Decision Letter 0]

4 Oct 2024

PONE-D-24-35489Repellency and toxicity of commonly used long-lasting insecticide-treated bed nets (LLINs) to bed bugsPLOS ONE

Dear Dr. Schal,

Thank you for submitting your manuscript to PLOS ONE. After careful consideration, we feel that it has merit but does not fully meet PLOS ONE’s publication criteria as it currently stands. Therefore, we invite you to submit a revised version of the manuscript that addresses the points raised during the review process.

We look forward to receiving your revised manuscript.

Kind regards,

Rajib Chowdhury, M.Sc.; MPH

Academic Editor

PLOS ONE

Journal Requirements:

2. Thank you for stating the following financial disclosure: This research was funded in part by grants from the Deployed Warfighter Protection Research Program, Department of the Army, U.S. Army Contracting Command, Aberdeen Proving Ground, Natick Contracting Division, Ft. Detrick MD (W911QY1910011), the U.S. Department of Housing and Urban Development Healthy Homes program (NCHHU0053-19), a Southern Region IPM Program grant (416682), and the Blanton J. Whitmire Endowment. 

3. Please note that your Data Availability Statement is currently missing [the repository name and/or the DOI/accession number of each dataset OR a direct link to access each database]. If your manuscript is accepted for publication, you will be asked to provide these details on a very short timeline. We therefore suggest that you provide this information now, though we will not hold up the peer review process if you are unable.

Additional Editor Comments:

Please carefully review the comments and suggestions from reviewers; and modify the manuscript accordingly.

Reviewers' comments:

Reviewer's Responses to Questions

**Comments to the Author**

1. Is the manuscript technically sound, and do the data support the conclusions?

Reviewer #1: Yes

Reviewer #2: Yes

2. Has the statistical analysis been performed appropriately and rigorously? 

Reviewer #1: Yes

Reviewer #2: Yes

3. Have the authors made all data underlying the findings in their manuscript fully available?

Reviewer #1: Yes

Reviewer #2: Yes

4. Is the manuscript presented in an intelligible fashion and written in standard English?

Reviewer #1: Yes

Reviewer #2: Yes

5. Review Comments to the Author

Reviewer #1: 1. The Title can be adapted as " Repellency and toxicity of various types of long-lasting insecticide-treated bed nets to bed bugs"

2. Abstract and Introduction started with malaria which does reflect well the research topic, these can be explained later in the text but not at the beginning.

3. Suggested to write the full name of DEET at least for the first time in the text.

4. Full form of IRB in line 90

5. Possible to rephrase the sentences from line 130 to 134 to make it more clear.

6. 12h to 96h or 12h till 96h?

7. For figure 2, 3 and 5, figure description is too long. Part of it can be included in text of the Result section.

8. Line 454, the proposed hypotheses could be common for all types of LLINs, described in the following paragraphs is good.

9. In discussion, 80% mortality in Fuller Mill strain exposed to all types of LLIN after 4 days is not clear from fig 5B as different survival ratios are shown after 96 hours.

Reviewer #2: Authors: Abstract: Line 15-16: Success of vector control programs hinges on community acceptance of products like long-lasting insecticide-treated nets (LLINs).

Suggestion: As you have not answered “Community acceptance of the product”, You should not raise the question.

Authors: Results: Line 209: Lack of olfactory repellency of LLINs

Suggestion: Word “Lack of” should not be used in the title, because it is your finding

Authors: Line 277: Marginal repellency of LLINs in contact assays

Suggestion: Word Marginal should not be used in the title, because it is your finding

Authors: Line 332: No olfactory attraction or repellency to chlorfenapyr

Suggestion: Similarly, word No should not be used.

6. PLOS authors have the option to publish the peer review history of their article (what does this mean?). If published, this will include your full peer review and any attached files.

Reviewer #1: No

Reviewer #2: **Yes: **Prof. Murari Lal Das

---

## [Editor Report · Decision Letter 1]

29 Oct 2024

Repellency and toxicity of long-lasting insecticide-treated bed nets (LLINs) to bed bugs

PONE-D-24-35489R1

Dear Dr. Schal,

We’re pleased to inform you that your manuscript has been judged scientifically suitable for publication and will be formally accepted for publication once it meets all outstanding technical requirements.

Kind regards,

Rajib Chowdhury, M.Sc.; MPH

Academic Editor

PLOS ONE
---

## [Editor Report · Acceptance letter]

1 Nov 2024

PONE-D-24-35489R1 

PLOS ONE

Dear Dr. Schal, 

I'm pleased to inform you that your manuscript has been deemed suitable for publication in PLOS ONE. Congratulations! Your manuscript is now being handed over to our production team.

Kind regards, 

on behalf of

Dr. Rajib Chowdhury 

Academic Editor

PLOS ONE